# Advance in the Management of Sepsis-Induced Coagulopathy and Disseminated Intravascular Coagulation

**DOI:** 10.3390/jcm8050728

**Published:** 2019-05-22

**Authors:** Toshiaki Iba, Jerrold H. Levy, Aditya Raj, Theodore E. Warkentin

**Affiliations:** 1Department of Emergency and Disaster Medicine, Juntendo University Graduate School of Medicine, 2-1-1 Hongo Bunkyo-ku, Tokyo 113-8421, Japan; aditya.raj@live.com; 2Department of Anesthesiology, Critical Care, and Surgery, Duke University School of Medicine, Durham, NC 27705, USA; jerrold.levy@duke.edu; 3Imperial College London, South Kensington, London SW7 2AZ, UK; 4Department of Pathology and Molecular Medicine, and Department of Medicine, McMaster University, 1280 Main Street West, Hamilton, ON L8S4L8, Canada; twarken@mcmaster.ca

**Keywords:** sepsis, disseminated intravascular coagulation, coagulopathy, antithrombin, thrombomodulin

## Abstract

Coagulopathy commonly occurs in sepsis as a critical host response to infection that can progress to disseminated intravascular coagulation (DIC) with an increased mortality. Recent studies have further defined factors responsible for the thromboinflammatory response and intravascular thrombosis, including neutrophil extracellular traps, extracellular vesicles, damage-associated molecular patterns, and endothelial glycocalyx shedding. Diagnosing DIC facilitates sepsis management, and is associated with improved outcomes. Although the International Society on Thrombosis and Haemostasis (ISTH) has proposed criteria for diagnosing overt DIC, these criteria are not suitable for early detection. Accordingly, the ISTH DIC Scientific Standardization Committee has proposed a new category termed “sepsis-induced coagulopathy (SIC)” to facilitate earlier diagnosis of DIC and potentially more rapid interventions in these critically ill patients. Therapy of SIC includes both treatment of the underlying infection and correcting the coagulopathy, with most therapeutic approaches focusing on anticoagulant therapy. Recently, a phase III trial of recombinant thrombomodulin was performed in coagulopathic patients. Although the 28-day mortality was improved by 2.6% (absolute difference), it did not reach statistical significance. However, in patients who met entry criteria for SIC at baseline, the mortality difference was approximately 5% without increased risk of bleeding. In this review, we discuss current advances in managing SIC and DIC.

## 1. Introduction

Activation of coagulation in sepsis is recognized as a host immune response against infection [1], however, over-activation of coagulation may be detrimental to the host [2]. Disseminated intravascular coagulation (DIC) is a result of disordered coagulation that is a laboratory and pathologic diagnosis, and a secondary response to an acute injury that has different underlying causes in addition to septic shock, and a definition that has changed over time [3]. In 2001, the International Society on Thrombosis and Haemostasis (ISTH) defined DIC as “an acquired syndrome characterized by intravascular activation of coagulation with loss of localization arising from different causes that can originate from and cause damage to the microvasculature, which if sufficiently severe, can produce organ dysfunction” [4]. This important definition better characterizes both the bleeding and the organ dysfunction that occurs. The term “DIC” initially included clinical hemorrhagic syndromes in patients with coagulopathy [5]. This type of consumptive coagulopathy is initiated by the release of multiple proinflammatory and procoagulant substances following tissue injury in patients with severe infections, trauma, obstetric emergencies, cardiogenic shock, and envenomation. At present, DIC conjures up different meanings in physicians, ranging from abnormal coagulation tests to ischemic limb gangrene and acrocyanosis [6]. Some clinicians consider DIC as “disseminated intracerebral confusion” as they have little understanding of this complex pathophysiologic process. As a result, we will review sepsis-associated DIC and of the newly proposed ISTH term, sepsis-induced coagulopathy (SIC).

## 2. Pathophysiology of Sepsis-Induced Coagulopathy and DIC

### 2.1. Activation of the Coagulation Cascade

Multiple factors activate coagulation during sepsis (Figure 1). Pathogen-associated molecular patterns (PAMPs) trigger inflammatory responses by activating multiple pathways and receptors that include pattern-recognizing receptors (PRRs) [7], damage-associated molecular patterns (DAMPs) such as high-mobility group box 1 (HMGB1), cell-free DNA, and histones. When these mediators are released from tissues and damaged cells, they can initiate inflammation and coagulopathy. Additional humoral mediators responsible for coagulopathy include proinflammatory cytokines such as interleukin (IL)-1, IL-6, tumor necrosis factor-α (TNFα), elastase, cathepsin G and complement system proteins, as a systemic response in sepsis [8]. In addition, neutrophil extracellular traps (NETs), consisting of DNA, histones and granule proteins released from activated neutrophils and other granulocytes, have been implicated in thrombus formation [9,10]. Recent research has also demonstrated interactions between NETs and extracellular vesicles, and NET-bearing extracellular vesicles that increase thrombin generation [11]. Tissue factor, a critical initiator of the extrinsic coagulation pathway, is expressed on macrophages, monocytes, and endothelial cells and plays a central role in the development of coagulopathy and DIC in sepsis [12] (Figure 2). Following cellular injury, tissue factor expressed on extracellular vesicles also further promotes procoagulant responses [13]. In summary, all of these inflammatory mediators activate thrombin as part of host defense responses to produce multiorgan failure that occurs in sepsis [14].

### 2.2. Fibrinolytic Shutdown

As initially described, the ISTH has defined DIC as systemic intravascular activation of coagulation arising from diverse causes, yet fibrinolysis is also profoundly suppressed in sepsis. Plasmin modulates fibrinolysis, and plasmin activity is regulated primarily by plasminogen activator (PA) and its inhibitor, plasminogen activator inhibitor-1 (PAI-1) (Figure 2 and Figure 3). Plasminogen activators convert plasminogen to the active enzyme plasmin through both pathways that include PA and contact activation. Vascular endothelial cells participate in the modulation of fibrinolysis by secreting both tissue-type plasminogen activator (t-PA) and PAI-1. Endothelial cell dysfunction is an essential feature in the pathogenesis of sepsis-associated DIC [15]. The marked increase in the PAI-1 level leads to disrupted fibrinolysis, and this key event represents the typical feature of thrombotic-type DIC. In addition to massive clot formation, impaired fibrinolysis prevents fibrin removal and leads to systemic microvascular thrombosis [16]. This occurs by fibrinolytic shutdown that is mainly induced by overproduction of PAI-1, and plasma levels of PAI-1 have been reported to be predictive of the severity and mortality in septic patients [17].

Other important pathways in coagulopathy/DIC include thrombin-activatable fibrinolysis inhibitor (TAFI). TAFI reduces plasmin generation and fibrin degradation by removing the plasminogen-binding-site from degraded fibrin after activation by thrombin or plasmin. In contrast to PAI-1, TAFI levels decrease due to extensive activation and consumption in sepsis, and therefore, may have less impact as compared to PAI-1 in the suppression of fibrinolysis [18]. In addition to PAI-1 and TAFI, decreased plasminogen levels may also contribute to decreased fibrinolytic activity [19]. Although these biomarkers are not standard laboratory tests, we believe that for potential clinical studies, it would be useful to monitor t-PA, PAI-1, TAFI, and plasminogen levels when defining coagulation/fibrinolysis balance in septic patients [20,21].

### 2.3. Endothelial Dysfunction

Vascular endothelium is the major target in sepsis, and one of the unique features of sepsis-associated DIC is endothelial injury [22] (Figure 1). In nonseptic DIC such as pregnancy-related and gestational DIC, amniotic fluid embolism and placental abruption, endothelial damage occurs due to secondary causes. In placental abruption, tissue factor and other procoagulant factors released from the placenta enter the maternal circulation and trigger coagulopathy, followed soon after by massive bleeding as a result of consumptive coagulopathy [23]. However, suppression of fibrinolysis in sepsis-associated DIC does not routinely occur in nonseptic DIC. In contrast, endothelial damage caused by microorganisms and inflammatory stimuli provoke transient t-PA release, followed by immediate upregulation of PAI-1 production, which contributes to massive fibrin deposition in the microcirculation [24,25]. Among the pregnancy-related complications, similar events to those involved in sepsis-associated DIC are seen in puerperal sepsis, eclampsia, and the HELLP syndrome.

In addition to the changes in fibrinolytic function, prothrombotic effects occur due to endothelial dysfunction due to decreased release of nitric oxide and prostacyclin, upregulation/expression of tissue factor and von Willebrand factor (VWF), and loss of the glycocalyx. The glycocalyx is the gel-like thin layer that covers the vascular endothelium and is an important target of infection and inflammation [26]. The glycocalyx is composed of membrane-binding proteoglycans, glycosaminoglycan side chains, and plasma proteins such as albumin and antithrombin. The antithrombotic activity of antithrombin is significantly increased by binding to the heparan sulfate side-chains of the glycocalyx, and the stability of the glycocalyx is increased by the binding of antithrombin to maintain vascular homeostasis [27,28]. During sepsis, glycocalyx components are shed into the blood and represent clinically relevant biomarkers, and loss of the glycocalyx is also thought to contribute to microcirculatory dysfunction [29]. 

### 2.4. Platelet Aggregation

Thrombocytopenia is also an important finding for diagnosing DIC in sepsis, in addition to coagulation abnormalities, as platelet count declines occur in virtually all patients with DIC. The platelet count in sepsis is affected by multiple factors. Despite increased levels of thrombopoietin, bone marrow production of platelets is often suppressed due to the effects of pathogenic toxins and inflammatory mediators. In sepsis, platelet activation occurs that contribute to platelet count decline, thrombin generation, inflammation, and VWF secretions [30]. Platelets are also activated by thrombin and inflammatory mediators including complement, and actively participate in the pathogenesis of sepsis-associated DIC (Figure 1). Thrombin induces platelet activation by cleaving of protease-activated receptors (PARs) expressed on the platelets to release platelet granule contents, such as ADP and serotonin [31]. Platelet activation through the PAR receptor activation is also associated with the generation of thromboxane A2 and release of a diverse array of proinflammatory cytokines [32]. Platelets also release HMGB1, one of the key DAMPs in sepsis, which upon activation, appears to play a critical role in thrombosis, monocyte recruitment, and production of NETs. However, attempts to inhibit platelet activation have not yielded favorable results in clinical trials and have failed to improve survival [33,34]. In contrast, the favorable effect of acetylsalicylic acid in patients with sepsis has been reported. Ouyang et al. [35] performed a meta-analysis on antiplatelet drugs and subgroup analysis showed that acetylsalicylic acid effectively reduced the ICU or hospital mortality. Overall, the effect of antiplatelet therapy is still inconclusive, while clinical studies evaluating thrombopoietin production for the prevention of organ damage in sepsis is currently under way [36].

### 2.5. Impairment of Anticoagulant Systems

Thrombin is a key mediator of the pathogenesis of sepsis-associated DIC, and antithrombin is the important physiological inhibitor of thrombin [37,38]. Antithrombin is estimated to provide 80% of the inhibitory activity against thrombin and other coagulation factors (Figure 2). Antithrombin inhibits multiple coagulation factors, including Factors X, IX, VII, XI and XII [38,39] (Figure 2). However, antithrombin levels are known to be reduced in sepsis because of consumption, impaired synthesis (especially in the subgroup of patients with sepsis shock-associated acute ischemic hepatitis [“shock liver”]) [6], extravasation, and degradation by elastase released from activated neutrophils [40,41]. Among these mechanisms, leakage from the vasculature is an important cause [42]. Recent discussions on the mechanisms of action have focused on the interaction between antithrombin and the glycocalyx on the vascular endothelial cells [43]. Antithrombin exerts local anti-inflammatory effects on the endothelial cell surface by binding to glycosaminoglycans. The healthy vascular endothelium is coated by a glycocalyx, however, its loss in sepsis increases the thrombogenicity and increases capillary permeability and cellular adhesion of the damaged endothelium [44]. Antithrombin is known to infiltrate the glycocalyx, bind to glycosaminoglycans, and thereby act to preserve the glycocalyx in sepsis [45]. Therefore, repletion of antithrombin exerts protective effects by inactivating the cytotoxic thrombin, but also through maintenance of the endothelial function by glycocalyx binding.

Another important anticoagulant system is the thrombomodulin/protein C system. Protein C pathways serve to maintain the balance between hemostasis and the host defenses in response to infection. Protein C is activated by thrombomodulin on the vascular surface, and activated protein C is known to exert multiple biologic activities, including antithrombotic, cytoprotective, and anti-inflammatory effects, to maintain vascular integrity. [46]. Activated protein C exerts antithrombotic activity through proteolytic inactivation of factors Va and VIIIa, while the cytoprotective activities are mediated by the effects on endothelial cells via the mediation of receptors such as endothelial protein C receptor (EPCR) and PAR-1. These multiple activities include anti-apoptotic activity, anti-inflammatory activity, regulation of gene expression, and stabilization of the endothelial barrier [47]. Derangement of this thrombomodulin/protein C system in sepsis is well known, and decreased protein C level is recognized as a useful biomarker of severe sepsis. Similar to antithrombin, protein C levels are especially vulnerable to depletion with acute ischemic hepatitis [6], and levels of both anticoagulant proteins have been reported to be significant predictors of mortality [48]. Thrombomodulin is proteolytically degraded from the cellular surface and circulates as a soluble form. Increased plasma levels of soluble thrombomodulin have been reported in patients with sepsis and organ dysfunction. Circulating thrombomodulin is expected to inhibit the adhesion of leukocytes to the endothelium since the serine/threonine-rich domain of thrombomodulin expressed on the vascular endothelial cells are reported to bind the leukocyte β2 integrins (LFA-1 and Mac-1) [49]. Although soluble thrombomodulin is eliminated by renal mechanisms, levels correlate with the severity of organ dysfunction or endothelial damage and have been measured to assess endothelial injury in sepsis [50].

## 3. Diagnosis of Sepsis-Induced Coagulopathy and DIC

### 3.1. Diagnostic Criteria for Sepsis-Associated DIC

As previously noted, the concept of DIC and approaches to the management differ significantly among clinicians, countries, and even institutions [51]. As a result, DIC is also referred to as “Disseminated International Confusion”. We believe it is important to define DIC, and the ISTH DIC Scientific Standardization Subcommittee (SSC) has published diagnostic criteria for overt DIC [51] (Table 1). The ISTH criteria for overt DIC use a scoring system based on the combined results of coagulation tests since no single test is adequate for the diagnosis. Other criteria have also been proposed to help identify patients at risk for DIC and to confirm the diagnosis of DIC. The ISTH criteria were designed based on the definition of DIC and the score has been shown to correlate with disease severity in sepsis [52]. Umemura et al. [53] reported that screening septic patients using specific DIC criteria can reduce mortality; however, there is no gold standard for diagnosing DIC. We believe it is important to understand the significance of biomarker abnormalities.

For determining diagnostic criteria for sepsis-induced DIC, specific considerations are as follows. The scoring system should be composed of readily available laboratory tests, performed serially, including platelet counts, prothrombin time-INR, fibrinogen, and fibrin-specific markers (e.g., fibrin D-dimers, soluble fibrin, and/or fibrin/fibrinogen degradation products). Although the ISTH criteria for overt DIC have been used as the global standard, one reported drawback is the delay in the diagnostic timing [54]. Yamakawa et al. [55] reported the importance of early intervention in a simulation model using data from 2663 sepsis patients, and reported the optimal cutoff score was an ISTH overt DIC score of 3 for determining the reduction of the mortality with anticoagulant therapy. This result suggests that anticoagulant therapy should be initiated even before the diagnosis of overt DIC.

### 3.2. Diagnostic Criteria for Sepsis-Induced Coagulopathy (SIC)

Coagulopathy is defined as “a condition in which the ability of the blood to clot is impaired” [56], a term that includes the thrombotic state in sepsis. In DIC, derangement of hemostasis and hypercoagulability can exist simultaneously, and patients can present with abnormal coagulation testing even if they are asymptomatic. For most clinicians, determining or managing laboratory-based coagulation abnormalities in critical care patients is confusing. However, in septic patients delays in interventions can be detrimental [55,57]. As a result, the concept of “sepsis-induced coagulopathy (SIC)” has been advocated by the ISTH DIC SSC [58]. According to this definition of SIC, evidence of organ dysfunction is assessed according to the SOFA scoring that includes respiratory, cardiovascular, hepatic, and renal dysfunction, as well as coagulopathy based on thrombocytopenia and a prolonged prothrombin time ratio (Table 1). In SIC scoring, SOFA score is defined in accordance with the revised definition of sepsis in reported in 2016 [59]. A comparison of the ISTH scoring system for overt DIC with the SIC scoring system in patients with sepsis showed that the SIC score was more sensitive than the ISTH criteria to detect coagulopathy [60].

### 3.3. Viscoelastic Testing for Sepsis-Associated DIC

In contrast to the use of the scoring system, viscoelastic testing has been studied to detect sepsis-associated DIC/coagulopathy. The two common methods of viscoelastic testing include thromboelastography (TEG^®^) and rotational thromboelastometry (ROTEM^®^). Both systems use whole blood with different activators, and both tests provide graphical and numerical indicators of clot initiation, formation, and lysis [61]. These point-of-care devices and are commonly used in trauma centers and surgical ICUs to detect coagulopathy [62]. Although their usefulness has not been tested in any large cohort of sepsis patients, good sensitivity and prognostic value have been reported in sepsis [63]. A hypercoagulability profile at admission has been reported to be helpful for the early detection of sepsis, while the degree of hypocoagulability may be associated with the severity [64]. Also, a lower maximum lysis was reported to predict a greater severity of organ failure [64], an important consideration since PAI-1 measurements are is not readily available.

### 3.4. Waveform Analysis of Clot Formation

The light transmittance waveform analysis on coagulation assays such as the prothrombin time (PT) test and activated partial thromboplastin time (APTT) test is another potential method for early detection of DIC [65]. Analysis and characterization of the PT and APTT tests on photo-optical coagulation analyzers provides additional information as to the clotting time, including the biphasic APTT waveform analysis for DIC [66]. As the degree of waveform abnormality correlated with3the severity of the hemostatic dysfunction and may provide useful information for determining the optimal timing for therapeutic interventions in DIC [67].

## 4. Differential Diagnoses of Sepsis-Induced Coagulopathy and DIC

Thrombocytopenia is often seen in sepsis and DIC; however, other diseases can also cause thrombocytopenia. Thrombotic microangiopathy (TMA) and heparin-induced thrombocytopenia (HIT) are probably the most important potential causes in critically ill patients. TMA includes thrombotic thrombocytopenic purpura (TTP) and hemolytic uremic syndrome (HUS), and is also characterized by thrombocytopenia and hemolysis. In TTP, platelet aggregation induced by the deficiency or inhibition of a disintegrin and metalloproteinase with a thrombospondin type 1 motif, member 13 (ADAMTS13) is the fundamental abnormality. ADAMTS13 is a specific protease that cleaves VWF multimers (to facilitate adhesion of platelets to injured endothelium), and severe depletion of ADAMTS13 is the hallmark of TTP [68].

Hemolytic uremic syndrome (HUS) is clinically characterized by intravascular hemolysis, thrombocytopenia, and acute kidney failure, and is categorized as Shiga toxin-producing *Escherichia coli* (STEC)-HUS, atypical HUS, or secondary HUS. The pathogenetic mechanism of STEC-HUS is a toxin-triggered endothelial injury, while that of atypical HUS is associated with genetic and/or acquired disorders of regulatory components of the complement system [69]. The common pathogenetic feature in STEC-HUS, aHUS, and secondary HUS is damage to the endothelial cells [70].

Secondary TMA is of diverse causation, such as pregnancy-related problems, collagen diseases, antiphospholipid syndrome, post-transplantation, malignancies, and certain drugs. Preeclampsia, eclampsia and the HELLP syndrome (hemolysis, elevated liver enzymes, low platelets) comprise pregnancy-related TMA, and the HELLP syndrome, a severe complication of pre-eclampsia, occurs in 0.2% to 0.8% of pregnancies. Its pathogenesis is not fully understood, but it is thought to be associated with inadequate placentation secondary to a maternal immune response to invading trophoblasts [71].

HIT can be classified as both an immune-mediated disorder as well as a consumptive thrombocytopenia; this is because the pathogenic anti-platelet factor 4 (PF4)/heparin antibodies activate platelets and trigger an associated procoagulant response [72]. Consequently, HIT is highly prothrombotic, with at least half of affected patients developing thrombosis, either arterial, venous, or microvascular, including a high frequency (~5%) of limb ischemic necrosis, including DVT-associated venous limb gangrene [73]. HIT requires treatment with a non-heparin anticoagulant, and attempting to distinguish HIT from non-HIT thrombocytopenia (prior to obtaining results of tests for HIT antibodies) is facilitated by use of a pretest probability scoring system, the 4Ts (thrombocytopenia, timing of onset, thrombosis, and other causes of thrombocytopenia) [73]. HIT is a preventable adverse drug reaction, as the frequency of HIT is approximately tenfold lower with low-molecular-weight heparin (LMWH) than with unfractionated heparin [74]. A consensus definition of HIT [75] emphasizes that diagnosis ideally requires both a compatible clinical picture plus a (usually) strong-positive PF4-dependent ELISA with corroborating laboratory evidence for platelet-activating antibodies. Almost all (>99%) cases of HIT are triggered by proximate (closely preceding) heparin exposure; however, so-called “spontaneous HIT syndrome” (with no preceding heparin exposure) is increasingly reported [76,77], and is now classified as an “autoimmune HIT” (aHIT) disorder [78]. Other aHIT disorders—which are characterized by HIT antibodies with both heparin-dependent and heparin-independent platelet-activating properties—include delayed-onset HIT, persisting HIT, HIT triggered by heparin “flushes” or fondaparinux administration, as well as the aforementioned spontaneous HIT syndrome [78]. Some patients with aHIT evince severe thrombocytopenia with overt DIC (elevated PT-INR, hypofibrinogenemia) and microthrombosis, thereby mimicking certain non-HIT disorders. For example, critically ill patients with septic or cardiogenic shock can sometimes develop bilateral lower-limb (and sometimes also upper-limb) peripheral ischemic limb necrosis despite presence of arterial pulses; this clinical picture of “symmetrical peripheral gangrene” is explained by circulatory shock complicated by (non-HIT) DIC and preceding “shock liver” (leading to depletion of natural anticoagulants, protein C and antithrombin) [79]. Since heparin can be appropriate for treating (non-HIT) DIC but is contraindicated in HIT, prompt discrimination is important, as recently described by the ISTH DIC SSC in their guidance for the differential diagnosis [80].

## 5. Treatment of Sepsis-Induced Coagulopathy and DIC

### 5.1. Unfractionnated Heparin and Low-Molecular-Weight Heparin

The fundamental strategy for sepsis-associated DIC management is treatment of the underlying infection [81]. Unfractionated heparin and low-molecular-weight heparin are the most commonly used and readily available anticoagulants for a variety of thromboembolic diseases. The efficacy of heparin and heparinoids for the treatment of DIC has previously been examined in clinical studies, however, their effectiveness and safety remain under debate (Table 2). A meta-analysis of 9 trials demonstrated that in most patients with sepsis, heparin therapy does not reduce organ injury or mortality, but was associated with an increase in the risk of bleeding [82].

A meta-analysis from China reporting the safety and efficacy of low-molecular-weight heparins (LMWHs) in patients with sepsis reduced the severity of sepsis and improved the survival but increased the incidence of bleeding events [83]. Regarding the usefulness of danaparoid sodium, a synthetic low-molecular-weight heparinoid, one small-sized randomized controlled trial (RCT) from Japan reported a favorable, although not statistically significant, effect. Heparin’s efficacy may be related to potential immunomodulatory and protective effects on the glycocalyx [84].

### 5.2. Antithrombin

The therapeutic approach to sepsis-associated DIC varies widely among different countries. Antithrombin concentrate and recombinant thrombomodulin are the most popularly used anticoagulants in Japan, whereas the these anticoagulants are not routinely used in most other countries for SIC/DIC. Indeed, global sepsis guidelines recommend against the use of antithrombin, since it reportedly increases the risk of bleeding [87]. This recommendation is based on the result of KyberSept, a large-scale phase III trial conducted to examine the effects of high-dose antithrombin. However, caution is needed, because this study was performed in patients with sepsis irrespective of whether DIC was present or not [88]. Wiedermann [89] performed a meta-analysis of its use in patients with sepsis and DIC and reported a beneficial effect of antithrombin on the mortality (risk ratio, 0.85; 95% confidence interval, 0.69 to 0.99; *I*^2^ = 0%). Based on similar results obtained from a meta-analysis in patients with sepsis-associated DIC, the Japanese Clinical Practice Guidelines for Management of Sepsis and Septic Shock recommend the use of antithrombin for DIC patients with decreased antithrombin activity [85]. Yatabe et al. [90] conducted a network meta-analysis to compare the effects and adverse events of antithrombin, recombinant thrombomodulin, heparin, and synthetic protease inhibitors in a total of 1340 patients. Although there were no significant differences in the risk of death or bleeding complications between placebo and any of the four anticoagulant treatment groups, the results also indicated that the use of antithrombin was associated with a fivefold higher likelihood of DIC resolution as compared to that in the placebo control group.

Antithrombin is often used with heparin since the anticoagulant activity of antithrombin is greatly accelerated by its binding to heparin. However, clinically, a considerable increase in bleeding risk with a possible reduction of its mortality benefit effect has been reported [91]. Physiologically, antithrombin binds to heparan sulfate on the endothelial surface, thereby contributing to local antithrombogenicity. Based on subanalysis of KyberSept data, intravenous heparin appears to alter the protective effect of antithrombin [92]. Thus, antithrombin should be administered without concomitant heparin administration.

### 5.3. Recombinant Activated Protein C

The thrombomodulin-protein C system is another important physiological anticoagulant mechanism (Figure 2). Reduced synthesis of protein C and its cofactor, protein S, in addition to impaired expression of thrombomodulin and EPCR on the endothelial cells, result in reduced activation of protein C, which is important for thrombin regulation. Activated protein C proteolytically degrades coagulation factors Va and VIIIa, thereby producing an antithrombogenic effect. Based on the above rationale, recombinant activated protein C (drotrecogin α) was developed, studied in a phase III study, and approved as a novel therapy for sepsis [93]. However, subsequent clinical trials failed to show a subsequent improvement of mortality and raised concern about bleeding [94]. As with any anticoagulant, increased bleeding events is likely a result of the antithrombotic and profibrinolytic effects of activated protein C, as activated protein C can induce clot lysis and increase bleeding [95]. As a result, drotrecogin α was withdrawn from the market by the manufacturer [96]. Subsequently, a recombinant activated protein C variant with normal signaling via EPCR and PAR1, but low anticoagulant activity was developed. A preclinical study reported reduced mortality in an animal model of sepsis, but the potential usefulness of this therapeutic approach has not been evaluated in patients [97]. 

### 5.4. Recombinant Thrombomodulin

The efficacy of recombinant thrombomodulin (ART-123, Asahi Kasei Pharma, Tokyo, Japan) has been studied in Japan including a phase III trial that reported better resolution of DIC compared to heparin [98]. A subgroup analysis in patients with sepsis-associated DIC revealed a reduction in the absolute mortality rate by 11.2% (ART-123: 21.4% vs. heparin: 31.6%; 95% CI: −9.1–29.4%) [99]. Following this study, a phase IIb trial was conducted outside of Japan that showed a reduction in the mortality rate by 3.8% that was not however statistically significant [100]. Most recently, a multi-national phase III trial in 800 patients with sepsis with organ dysfunction and coagulopathy (platelet count < 150 × 10^9^/L and PT-INR > 1.4) reported a reduction of the mortality rate by 2.6%, which was again not statistically significant; more than 20% of the patients recovered before the initiation of the treatment in this study, and the mortality rate reduction was greater (approximately 5%) among patients who had fulfilled the entry criteria at baseline [101].

In a meta-analysis of all the ART-123 trials, Yamakawa et al. [102] reported a reduction of the mortality rate by 13% (relative risk [RR]: 0.87, 95% CI, 0.74–1.03, *p* = 0.10). In contrast to activated protein C, ART-123 is reported to be associated with a lower risk of bleeding potentially due to thrombomodulin’s antifibrinolytic activity unlike activated protein C and is less likely to cause bleeding. Further, circulating levels of activated protein C activity did not increase after the administration of ART-123 [86]. These findings have supported the use of recombinant thrombomodulin for the treatment of sepsis-associated DIC or SIC in Japan.

### 5.5. Recombinant Tissue Factor Pathway Inhibitor

Tissue factor pathway inhibitor (TFPI), a Kunitz-type serine protease inhibitor, is another important anticoagulant system. TFPI modulates the coagulation system by binding directly to the tissue factor-Factor VII/Factor VIIa complex and Factor Xa [103] (Figure 2). Since tissue factor plays a key role in sepsis-associated DIC, its inhibition is thought to have a beneficial effect. Regarding the use of recombinant TAFI (tifacogin), two RCTs, one targeting patients with sepsis and another targeting patients with pneumonia, have been performed. Although recombinant TFPI was studied in septic patients, it failed to show any mortality benefit [104]. However, a *post hoc* analysis suggested a trend toward improvement of the survival among treated patients with community-acquired pneumonia, the second RCT targeted patients with pneumonia [105]. The results showed similar mortality rates between the tifacogin group and the placebo group. Interestingly, the treatment group in the recombinant TFPI RCT showed a greater reduction in the levels of prothrombin fragment 1+2 and thrombin-antithrombin complex, suggesting that the biological activity *per se* is not responsible for the overall benefit. These findings also suggest the potential role of inhibiting the proinflammatory effects is an important effect using anticoagulants. Of note is that these two studies were performed in patients with sepsis or pneumonia, and not in patients with coagulopathy as defined by DIC, a consistent finding of trials that failed using coagulation based therapies, and not targeting patients of sepsis with coagulopathy.

## 6. Summary

Clinicians have often considered DIC to be an abbreviation for “Death Is Coming” due to the absence of any established treatment. With growing concerns regarding sepsis and potential therapies to reduce mortality, differences in the diagnosis and management of sepsis-associated DIC have hindered advances in the understanding/management of this condition. We believe that targeted therapy using precision medicine, namely anticoagulants, is an important consideration in sepsis but will not be effective unless the patient has concomitant DIC. Early recognition of the coagulation disorder and prompt initiation of targeted anticoagulant therapy may alter outcomes of sepsis, but early and rapid diagnosis is important for this therapeutic consideration.

## Figures and Tables

**Figure 1 jcm-08-00728-f001:**
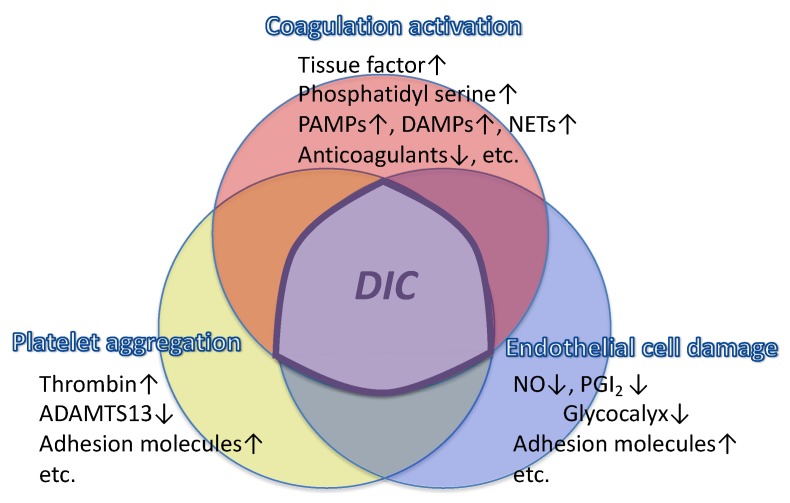
Three major factors contribute to DIC that include coagulation activation, platelet aggregation, and endothelial damage. Tissue factor expressed on the leukocytes and phosphatidylserine on the damaged cell membrane activate coagulation, decreased physiologic anticoagulant systems accelerate clot formation, and platelet aggregation is stimulated by thrombin and other inflammatory mediators. Endothelial damage reduces the antithrombotic milieu of the vascular lumen. PAMPs: pathogen-associated molecular patterns, NETs: neutrophil extracellular traps, DAMPs: damage-associated molecular patterns, ADAMTS13: a disintegrin and metalloproteinase with a thrombospondin type 1 motif, member 13, NO: nitric oxide, PGI_2_: prostagrandin I_2_, DIC: disseminated intravascular coagulation.

**Figure 2 jcm-08-00728-f002:**
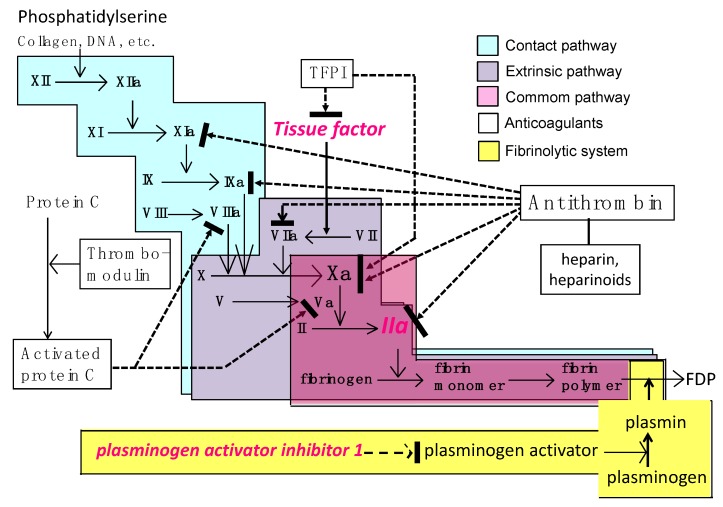
The coagulation, fibrinolytic, and natural anticoagulant systems. Both the extrinsic pathway and contact pathway are activated in sepsis. Tissue factor, expressed on the monocytes, endothelial cells, and extracellular vesicles triggers the extrinsic pathway, while the phosphatidylserine residue present in various cell membranes initiates the contact pathway of coagulation. Antithrombin/heparin, thrombomodulin/protein C, and tissue factor pathway inhibitor (TFPI) are the three major physiologic anticoagulant systems. FDP: fibrin/fibrinogen degradation products.

**Figure 3 jcm-08-00728-f003:**
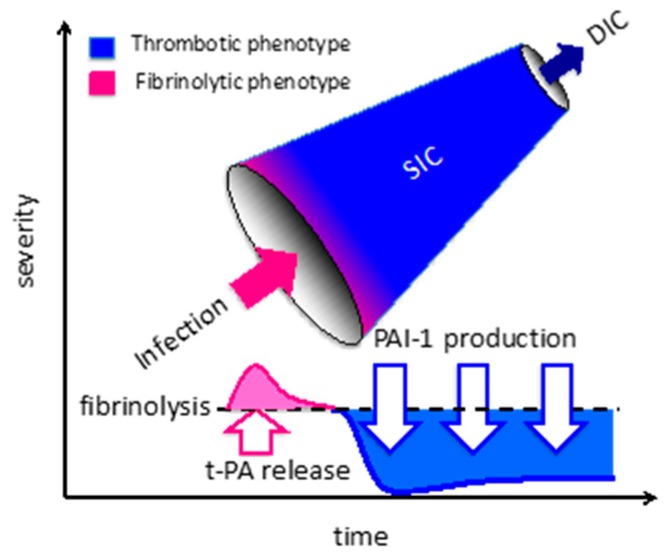
The sequential change from sepsis-induced coagulopathy to disseminated intravascular coagulation. Sepsis-induced coagulopathy progresses to disseminated intravascular coagulation (DIC) if the infection/inflammation is severe enough. Transient activation of fibrinolysis is observed initially, due to the release of tissue-type plasminogen activator (t-PA) from the vascular endothelial cells. Subsequently, the fibrinolytic system is suppressed by the production of plasminogen activator inhibitor-1. The imbalance between coagulation and fibrinolysis leads to a hypercoagulable state and organ dysfunction in sepsis.

**Table 1 jcm-08-00728-t001:** ISTH overt DICand SIC scoring systems.

		ISTH overt DIC	SIC
Item	Score	Range	Range
Platelet count (×10^9^/L)	2	<50	<100
	1	≥50, <100	≥100, <150
FDP/D-dimer	3	Strong increase	-
	2	Moderate increase	-
Prothrombin Time (PT ratio)	2	≥6 sec	(>1.4)
	1	≥3 sec, <6 sec	(>1.2, ≤1.4)
Fibrinogen (g/mL)	1	<100	−
SOFA score	2	-	≥2
	1	-	1
Total score for DIC or SIC		≥5	≥4

ISTH: International Society on Thrombosis and Haemostasis; DIC: disseminated intravascular coagulation; SIC: sepsis-induced coagulopathy; SOFA: sequential organ failure assessment. SOFA score: score is the sum of 4 items (respiratory SOFA, cardiovascular SOFA, hepatic SOFA, renal SOFA).

**Table 2 jcm-08-00728-t002:** Therapeutic agents for DIC/SIC.

Agent		Recommendation	Rationale
Unfractionated heparin, low-molecular-weight heparin		None, except for deep vein thrombosis prevention	Efficacy for venous thromboembolic prophylaxis is expected, but insufficient supportive data for DIC treatment.
Antithrombin	If it is available	Optional choice	Although phase III study (KyberSept) did not find efficacy in sepsis (but did show increased bleeding), meta-analysis of DIC patient subset showed beneficial effect in survival [85].
	If it is not available	None	-
Recombinant thrombomodulin	If it is available	Optional choice	Although phase III study (SCARLET) did not show efficacy in sepsis with coagulopathy, meta-analysis found trend toward improved survival [86].
	If it is not available	None	-

DIC: disseminated intravascular coagulation, SIC: sepsis-induced coagulopathy.

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
