# Peer review of "Advance in the Management of Sepsis-Induced Coagulopathy and Disseminated Intravascular Coagulation"

_jcm, 2019, doi:10.3390/jcm8050728_

Reviewer 1 Report

The manuscript entitled “Management of sepsis-induced coagulopathy and disseminated intravascular coagulation” represents a comprehensive review article about a fairly new entity on sepsis-induced coagulopathy.

The topic is of clinical interest and appeals to the readership in critical care medicine.

Authors reviewed all the important sections for this condition:

·         Pathophysiology: They included all the mechanisms that are involved in DIC and SIC. For each mechanism, they wrote in detail.

·         Diagnosis: Authors organized it to gather all the possible diagnostic workups. However, it is better to include Viscoelastic testing for sepsis-associated DIC & Waveform analysis of clot formation under this section.

·         Differential diagnosis: all the important differentials are included.

·         Treatment:  All the treatment options were discussed.

Overall, it is a comprehensive review article that covers all the required information, if not more, on SIC.

Author Response

The authors appreciate the reviewer for the supportive comments. The ‘Diagnosis’ section was rearranged accordingly. Thank you again for your kind suggestion.

Reviewer 2 Report

This is a timely and well written review. In the light of several views over diagnosis and treatment of DIC and sepsis like conditions this review provides a balanced account of current literature. Also the recommendations and opinions provided in the review are not over-reaching. The section on the coagulation and fibrinolytic system is particularly well written. The review has  one specifici (Figure 1) , one general (Figure 2) figure. The review might benefit from a few more  figures illustrating the functional connections of the coagulation/fibrinolytic system to the clinical phenotypes. The Figure 2 can be individually addressed in detail in the respective sections following it. However, this is just a minor comment and the current Figures also adequately address the issue at hand. Diagnostic section is aided by adequate and informational tables. I recommend a minor review for typographical errors and grammatical inconsistencies.

Author Response

The authors appreciate the reviewer for the supportive comments. We added Fig. 1 and its legend accordingly. We apologize for the typographical errors and grammatical insistencies. They were corrected in the text.

Reviewer 3 Report

This is a comprehensive, balanced, and critical review of septic coagulopathy and disseminated intravascular coagulation (DIC), focusing on recent advances in management. Extensive recent literature was reviewed, and critically discussed. Novel insight and practical recommendations was provided. The text demonstrated good scientific accuracy and interest to the broader clinical and research community. I have only minor comments:

The title should be more accurately to include clauses such as "latest advance in" or similar phrases.

Fig 2 should have arrows on both x and y axes.

Fig 2, the color of common pathway should be separated from intrinsic/extrinsic pathway colors.

Please revise lines 278-299 as the text substantially overlap with several existing publications.

Author Response

The authors appreciate the reviewer for the supportive comments. In the following part, we show the itemized reply to each comment.

The title was changed to“Advances in the management of sepsis-induced coagulopathy and disseminated intravascular coagulation” accordingly.

We added the arrows on both x- and y-axes in Fig. 1.

The color of the common pathway was added in Fig. 2.

HIT section (lines 278-299) was entirely rephrased to the new descriptions.